# Restoration of Fertility in Patients with Spontaneous Premature Ovarian Insufficiency: New Techniques under the Microscope

**DOI:** 10.3390/jcm10235647

**Published:** 2021-11-30

**Authors:** Marie Mawet, Sophie Perrier d’Hauterive, Laurie Henry, Iulia Potorac, Frédéric Kridelka, Michelle Nisolle, Axelle Pintiaux

**Affiliations:** 1Service de Gynécologie-Obstétrique, Uliège, Site du CHU, Avenue de l’Hopital 1, 4000 Liège, Belgium; frederic.kridelka@chuliege.be (F.K.); axelle.pintiaux@gmail.com (A.P.); 2Service de Gynécologie-Obstétrique, Uliège, Site du CHR, Boulevard du 12ème de Ligne, 4000 Liège, Belgium; sperrierdh@gmail.com (S.P.d.); laurie.henry@chrcitadelle.be (L.H.); michelle.nisolle@chrcitadelle.be (M.N.); 3GIGA-Stem Cells, Uliège, Site du CHU, Avenue de l’Hopital 1, 4000 Liège, Belgium; 4Service d’Endocrinologie, Uliège, Site du CHU, Avenue de l’Hopital 1, 4000 Liège, Belgium; julia_potorac@yahoo.com

**Keywords:** primary ovarian insufficiency, premature ovarian insufficiency, fertility restoration, platelet-rich plasma, mesenchymal stem cells

## Abstract

Premature ovarian insufficiency (POI), a condition affecting up to 1% of women by the age of 40 years, is characterized by an extremely low chance of spontaneous pregnancy. Currently, fertility restoration options are virtually nonexistent for this population. To become pregnant, the only solution is egg donation. Interestingly, animal studies have provided encouraging results in terms of fertility restoration, and consequently, research has begun into the most promising approaches for women suffering from POI. The PubMed database was searched for studies in which techniques aiming at restoring fertility in women with spontaneous POI were tested. Although robust studies are lacking, the literature suggests a positive effect of certain techniques on fertility restoration in women with POI. The most promising approaches seem to be intraovarian injection of autologous platelet-rich plasma or of mesenchymal stem cells. In addition to these, in vitro and mechanical activation of dormant follicles and etiology-driven therapies have also been studied with mixed results. No safety concerns were raised in these studies. The absence of robust studies does not allow us to draw meaningful conclusions on the efficacy or superiority of any single technique at this stage, and so research in this area should continue using robust study designs, i.e., multicenter randomized controlled trials including sufficient subjects to achieve statistical power.

## 1. Introduction

Premature ovarian insufficiency (POI) was defined in 2016 by the European Society of Human Reproduction and Embryology (ESHRE) as the onset of menstrual disturbance (oligomenorrhea or amenorrhea) for at least 4 months, associated with serum follicle stimulating hormone (FSH) levels ≥ 25 IU/mL measured at least twice with a 4-week interval, before the age of 40 years [1]. 

Clinicians distinguish iatrogenic POI, which is the consequence of oncological treatments (chemo- and/or radiotherapy) or ovarian surgery performed during premenopausal stages, from spontaneous POI, whose etiology is often complex. It may be the consequence of different genetic mutations or of autoimmunity (5% of all spontaneous POI cases). However, in 90% of the cases, no etiology is found, a condition referred to as idiopathic POI [1]. Spontaneous POI occurs in 0.1% of women by the age of 30 years and 1% of women by the age of 40 years.

Approaches to preserve fertility in cases of iatrogenic POI have been developed and have already shown efficacy [2]. However, fertility restoration in women suffering from spontaneous POI is almost never successful. Oocyte donation, surrogacy, and adoption are often the only options to achieve motherhood but are not always accepted by the patients, who may express the desire for genetically related offspring. In addition, oocyte donation remains illegal in some countries [3]. 

Some research groups are experimenting with new strategies in the hope of re-activating the small pool of quiescent follicles that is still present in the ovaries of women with spontaneous POI [4,5]. The aim of this article is to describe different attempts at fertility restoration that have been conducted in the last decade.

## 2. Materials and Methods

We performed a PubMed search with the terms “premature ovarian insufficiency” or “premature ovarian failure” and “fertility treatment”, “fertility restoration”, or “fertility rejuvenation” with the “human” limit. 

The main criteria for the study selection were: interventional studies specifically evaluating strategy(ies) to restore fertility in a population of women suffering from spontaneous POI, and reports published in English from 1 January 2011 to 6 August 2021, with the exclusion of reviews and meta-analyses. 

Out of the 437 articles retrieved, 392 were excluded based on their title and/or abstract (Figure 1). After reading the full texts, 13 articles were found to meet the selection criteria. Initially, we wished to use the PRISMA guidelines for the report of their research results. However, due to both the limited number of studies retrieved and the nature of these studies, the guidelines were ultimately not applicable, and the data from the 13 articles of interest were collected into five sections presented below, starting with the better-documented techniques.

## 3. Results

### 3.1. Autologous Platelet-Rich Plasma Intraovarian Injection 

After centrifugation of a subject’s whole blood, it is possible to collect plasma that has high concentrations of platelets, called platelet-rich plasma (PRP), which may subsequently be reinjected into the subject at specific injured sites, a process referred to as autologous PRP injection. This process has been known for more than 50 years, and has previously demonstrated a certain degree of efficacy in some health conditions requiring tissue healing and regeneration, notably in orthopedics, cardiothoracic surgery, plastic surgery, dermatology, dentistry, and diabetic wound healing [6]. The mechanism of action of PRP is complex and remains incompletely understood: after activation, platelets release a series of cytokines, growth factors, and hormones, capable, among other actions, of attracting stem cells, enhancing angiogenesis, and activating cellular differentiation and proliferation [7]. Surprisingly, very few attempts at using PRP have been undertaken in the gynecological field until recently, when it was suggested as a means to improve pregnancy rates in some women with thin endometrial lining and recurrent implantation failures, and in some poor responders. These studies have been well-summarized by Sharara et al. in a 2021 review [6]. 

Sfakianoudis et al. were pioneers in studying the effects of PRP in women with POI. They initially reported, in two publications, the cases of four women presenting with POI who had rejected the option of oocyte donation [8,9]. The subjects, aged between 27 and 46 years, were amenorrheic for more than one year (range: 12 to 26 months). They agreed to undergo a transvaginal intraovarian injection of autologous PRP guided by ultrasonography. Within 2 months after this procedure, all four women resumed menses and became pregnant: conception was achieved naturally in three of them and using a natural in vitro fertilization (IVF) cycle in one of them. This last woman reported a spontaneous abortion at week 5 of gestation while the three other women completed healthy, complication-free pregnancies.

Following these encouraging results, the same research group conducted a pilot uncontrolled study in which 30 women diagnosed with POI underwent transvaginal intraovarian autologous PRP injection [10]. Recently, Cakiroglu et al. also reported the results of intraovarian injection of autologous PRP in 311 women diagnosed with POI, which currently constitutes the largest study on the topic [11]. The results of these two studies are displayed in Table 1. In both studies, the parameters reflecting the activity of the hypothalamic–pituitary–ovarian axis and of the ovarian reserve generally improved, although the results seemed variable among the participants. The total pregnancy rate in each study was 10.0% and 11.6%, respectively. It should be noted that, in the second study, a little more than 30% of the pregnancies were achieved using ovarian stimulation. 

Interestingly, in both studies, a lower FSH level and a higher AMH level prior to treatment were positively correlated with a better outcome of the technique. Therefore, the authors suggested that intraovarian PRP injection helps by stimulating existing pre-antral/early antral follicles. The presence of these follicles could thus be a prerequisite for conditioning the magnitude of the response. 

No safety concerns were reported in these studies by the authors.

### 3.2. Human Mesenchymal Stem Cells

Over the last 10 years, numerous preclinical studies conducted in animal models of POI have suggested the benefit of stem-cell-based therapy in restoring ovarian function. Most of these studies were conducted in rodents with chemotherapy-induced POI. Stem cells from different sources have been tested, but the most promising results were seemingly obtained with stem cells of mesenchymal origin (MSCs), which can notably be found in amniotic fluid, menstrual blood, the umbilical cord, bone marrow, and adipose tissue (Figure 2). In almost all of these animal studies, an improvement in ovarian function was recorded (a decrease in FSH and an increase in E2 levels) and, in the studies evaluating mating, treated animals showed higher pregnancy and live birth rates than controls [12].

The mechanism of action of MSCs in restoring damaged tissues is extremely complex and still under active investigation. Although beyond the scope of this article, based on current knowledge, we may summarize that MSCs’ repair potential relies on two properties: first, the differentiation potential, and second, the secretome [13,14]. The differentiation potential is the capacity of a given cell to transform into various types of cells and, consequently, to replace those damaged or absent in the target tissue. In line with this, MSCs were shown to differentiate into different ovarian cell lineages [15]. However, some evidence suggests a limited effect of this mechanism on the positive impact of cell therapy in damaged ovaries. Grady et al. notably showed in aged mares that the DNA of donor MSCs was not recovered in the treated ovaries [16]. More attention is now being given to the secretome, which consists of all the molecules secreted by MSCs into the extracellular space (cytokines, chemokines, and growth factors) and which plays an essential role in promoting angiogenesis, inhibition of apoptosis and immune response modulation. Figure 2 displays some of these molecules and their activity.

In addition to the large number of preclinical studies, only five clinical studies have been published to date on women with POI: three single-arm studies and two controlled studies (one of them randomized). These studies are summarized in Table 2 [17,18,19,20,21]. Different methods were used in the five studies: three different stem cell sources (bone marrow, umbilical cord, or adipose tissue), two routes of intraovarian injection (laparoscopic or transvaginal), and different injection sites (either in both ovaries, in one ovary, or in one ovary and one ovarian artery). No adverse reactions were reported in any of these four studies. Results in terms of pregnancy rate range from 0% to 14%. This heterogeneity in study designs and methodologies, associated with the limited sample sizes, precludes drawing meaningful conclusions at this time on the efficacy of stem cell therapies to restore fertility in women with spontaneous POI. There is, however, no doubt that much more data will be available in the near future since there were 23 studies on the topic registered in the clinicaltrial.gov database as of 12 August 2021.

### 3.3. In Vitro Activation of Dormant Follicles

The basic principle behind this method relies on in vitro activation of the dormant primordial follicles that persist in a certain percentage of women with spontaneous POI, followed by regrafting the activated ovarian tissue back into the subject. Before initiating a study in humans, Kawamura et al. demonstrated, using murine and human ovarian tissue, the possibility of activating primordial follicle growth in vitro by two different approaches. First, by fragmentating the ovarian cortex tissue, the Hippo signaling pathway (a cascade resulting in negative growth stimulation of the ovaries) is disrupted. Consequently, growth factors increase in the fragmented ovarian tissue, which promotes follicle development. The second approach consisted of incubating the ovarian tissue with two follicle-activating drugs, namely, an inhibitor of phosphatase with TENsin homology deleted in chromosome 10 (PTEN) and an activator of phosphatidynositol-3-kinase (PI3 kinase) [22,23]. The authors combined these two techniques to attempt fertility restoration in women with POI. 

The single-arm clinical study included 37 participants with POI who agreed to undergo laparoscopic removal of one (or both) ovary(ies) and subsequent re-implantation of fragmented ovarian tissues previously incubated with PI3 kinase and PTEN [5,24]. Follicular growth and hormonal levels were monitored weekly. In nine subjects, antral follicles were detected and follicle stimulation by FSH was started. Oocyte removal was possible in six patients, for a total of 24 oocytes retrieved. Four women had embryo transfer: chemical pregnancy was detected in three patients (pregnancy rate: 8.1%): one ended in a miscarriage and the two others in the delivery of two healthy babies (live birth rate: 5.4%). No adverse events were reported in these articles.

Before performing the in vitro activation, a small part of the ovarian cortex of all the participants was histologically analyzed to detect the presence of residual follicles. No follicles could be seen in 17 women, generally in those who had developed POI more than 5 years before the ovariectomy. In these women, after transplantation of the in vitro activated ovarian tissue, no antral follicle was detected (failure of the technique). This provides interesting information: the number of residual activatable follicles decreases over time after the cessation of menses; therefore, fertility restoration attempts should be started as soon as the diagnosis of POI is established in order to increase the success rate.

### 3.4. Mechanical Activation of Dormant Follicles

As described in the previous section, ovarian cortical fragmentation seems to stimulate follicle growth by disrupting the Hippo signaling pathway. It was hypothesized by Zhang et al. that performing local injury to the ovarian cortex could be sufficient to disrupt the Hippo signaling pathway [25]. They enrolled 80 women with spontaneous POI and, under laparoscopy, performed ovarian biopsies in the left ovary and three scratches in the right ovary. There was no control arm in the study. No safety issues were reported by the authors. Histological analysis of the ovarian biopsies revealed that among the 80 patients, only 12 (15.0%) still presented ovarian follicles at the time of the surgery. During the 6 months of follow up, ovarian function resumption was seen in 11 of these 12 women, while no sign of ovarian activity was seen in the 68 without follicles on the histological analysis. Following ovarian stimulation, 10 patients had oocyte retrieval, for a total of three mature oocytes retrieved. After in vitro fertilization, two women had embryos transferred, giving birth to one healthy child (pregnancy rate: 1.25%). 

Although somewhat disappointing, the results of this experimental study confirm the important information already suggested in other studies: the success of the techniques largely depends on the presence of existing follicles in the ovaries.

### 3.5. Immunomodulation in Autoimmune Forms

As mentioned above, in most cases, no underlying cause was found to explain POI. However, a certain percentage of women present autoimmunity directed against the ovarian tissue. One might therefore hypothesize that treating the underlying condition could improve ovarian activity. A few reports have suggested that treatment with corticosteroids may improve fertility in women with autoimmune POI. However, corticotherapy is not devoid of unwanted and potentially serious side effects [26]. Safer immunosuppressive treatments are currently available. The utility of such therapies in autoimmune POI is suggested in a case report published in 2011 by Ferrau et al., who reported a pregnancy that occurred soon after a woman suffering autoimmune POI was started on treatment with azathioprine 100 mg/day for ulcerative colitis. Since this interesting case report, to the best of our knowledge no other publication has been released on the topic.

## 4. Discussion

The goal of this work was to describe some innovative approaches in the last decade to restoring fertility in women presenting with spontaneous POI. It is important to recognize the interest in research in this field, where, currently, the only hope for the patients to become pregnant remains egg donation, a solution that is not always accepted by the woman or her partner. In addition, egg donation is not legal in some countries. In that case, adoption becomes the single solution that can be offered to these patients [3].

The ground for research aimed at restoring fertility in women with POI has been laid by studies conducted on animal models, which delivered promising results, opening the door to some hope for human medicine. In this last decade, a series of clinical studies have therefore been conducted in women presenting with POI. For the moment, the most studied techniques are intraovarian injection of PRP and use of MSCs. Some interest was also paid to mechanical techniques aimed at stimulating dormant follicles in the ovaries (in situ and/or in vitro).

Although these trials seem to deliver promising results, at this stage, the lack of robust studies prevents us from drawing meaningful conclusions on the relative efficacy or superiority of the various techniques. As shown in this article, the currently available clinical data were generally obtained through case reports or single-arm studies. Only two controlled studies were found, in which no sample size seems to have been calculated, which precludes them from delivering statistically significant data. However, it is known that ovulation has an approximately four percent chance per month to spontaneously occur in women with POI, as a consequence of which spontaneous pregnancies have been reported in these patients [27]. The phenomenon remains totally unpredictable, but should be considered when conducting interventional studies. Therefore, to demonstrate the superiority of any method aimed at restoring ovarian function in these women, a first important step would be to include a comparative placebo group in the studies. In addition to a placebo, another important step to improve our knowledge would be to compare in randomized controlled trials different methods for a single approach, (e.g., different routes of administration, different doses, different preparation conditions, etc.) and, once the most effective method has been delineated, to compare different approaches (e.g., intraovarian injection of PRP versus injection of MSCs).

Performing well-conducted randomized controlled studies appears to be crucial, and probably necessitates a large population of patients only achievable in multicentric studies. Large-scale international collaboration could be a good way to facilitate recruitment and to achieve, in a timely fashion, the required number of participants. We could also consider building an international database that would include all the POI patients of the participating centers, as has been performed to improve research on other rare diseases. The frequent lack of knowledge regarding the underlying causes of the disease is another difficulty encountered in this field. One might suspect that defining specific pathologic groups could help to identify the best approach depending on the POI cause. This is notably illustrated by the anecdotal case report showing fertility improvement in a woman with autoimmune POI treated with immunosuppressive agents. Here also, an international database could facilitate the recruitment of patients depending on their underlying pathology when the latter is known.

As shown in a couple of articles described above, one of the most important parameters that seems to condition response to treatment could be the presence of remaining follicles in the ovaries at the time of study. Histological analysis of the ovarian cortex seems therefore a good approach to selecting patients. Less-invasive approaches such as dosage of AMH could also be helpful, even though it seems to be of less importance [24]. Crucially, as suggested by Suzuki et al. and others, the pool of remaining follicles appears to decrease with time after the cessation of menses [24,25]. Patients should be informed of this, and attempts to restore fertility should start soon after the diagnosis of POI whenever possible.

No adverse events were reported in any of the studies reviewed in this article. It should however be noted that these techniques use transvaginal or laparoscopic ovarian manipulations, which are not devoid of undesirable side effects. In addition, some specific concerns linked to the clinical use of stem cells must be considered [28]. Among them, the possible tumorigenic effect of MSCs is probably the most significant one. With their ability to proliferate for a long period of time, their high viability, and their resistance to apoptosis, MSCs could be suspected to behave as tumor cells. In addition, some animal studies also suggested that MSCs could potentiate growth of pre-existing tumors in the host. In addition to this oncological concern, other safety issues are suspected: injection of MSCs appears to have prothrombotic effects and could also be a vector for prion or viral transmission. The current studies conducted in humans with MSCs are too limited in terms of sample size and duration to properly evaluate the significance of these safety concerns, but there is no doubt that the increasing interest around MSC-based therapy will help delineate their exact safety profile.

Finally, in this very early phase of clinical research, it is particularly important for researchers to report the results of their studies, including the less- or unsuccessful ones. Indeed, this represents an important way to advance research progress by avoiding the repetition of failed approaches. Therefore, unsuccessful studies are just as instructive as successful ones, and their publication should be encouraged by the scientific community.

## 5. Conclusions

Research in the field of ovarian function restoration in women with spontaneous POI is of high interest, but requires well-conducted, randomized controlled studies with sufficient statistical power, probably using multicentric international designs.

## Figures and Tables

**Figure 1 jcm-10-05647-f001:**
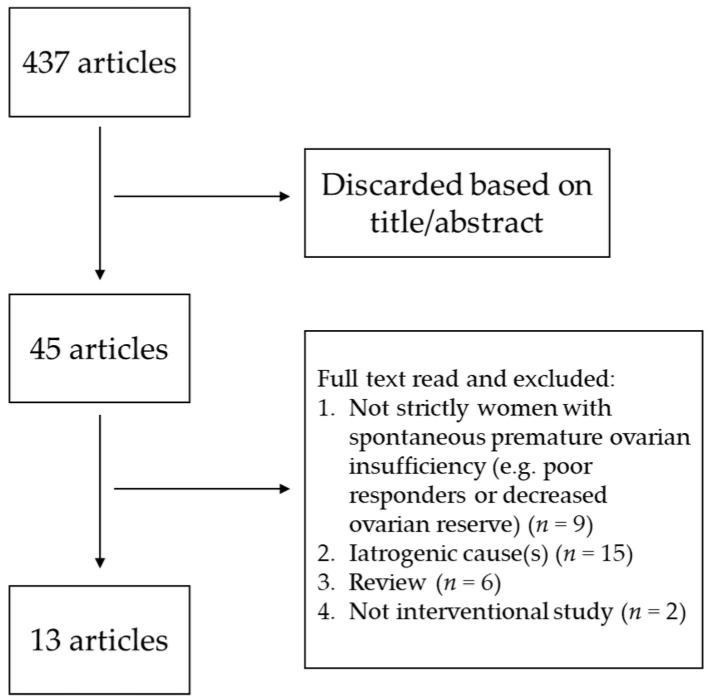
Flow diagram for the selection of the articles included in this review.

**Figure 2 jcm-10-05647-f002:**
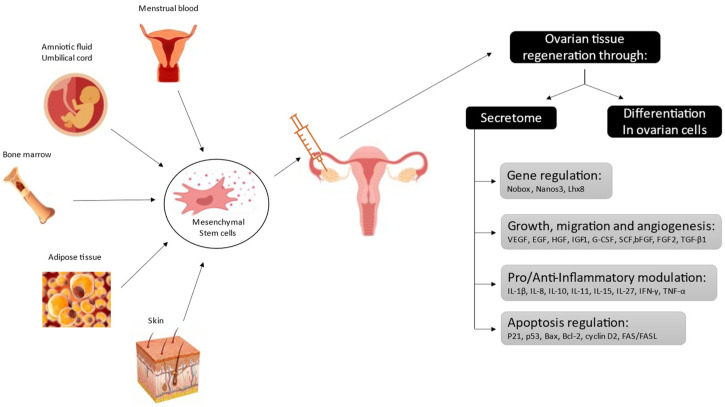
Origins of mesenchymal stem cells that have been tested in preclinical studies using premature ovarian insufficiency models, along with their potential mechanism of action in restoring fertility (adapted from Ahmadian S et al., 2020).

**Table 1 jcm-10-05647-t001:** Clinical studies conducted with intraovarian injection of autologous platelet-rich plasma to restore fertility in women suffering from spontaneous premature ovarian insufficiency.

Reference		Sfakianoudis K. et al., 2020				Cakiroglu Y. et al., 2020	
Number of patients Mean age (years)		*n* = 30 *				*n* = 311 34.8 ± 4.3	
		Success group (*n* = 18, 60.0%) 35.11 ± 1.57		Failure group (*n* = 12, 40.0%) 35.92 ± 1.93			
		Baseline	3 months after intraovarian injection of PRP	Baseline	3 months after intraovarian injection of PRP	Baseline	After intraovarian injection of PRP
Mean duration of amenorrhea (months)		10.06 ± 2.62	Return of menstruation	10.17 ± 4.76	No return of menstruation	>4	NA
Hypothalamic– pituitary–ovarian axis parameters (means)	FSH (UI/mL)	40.61 ± 6.05	20.67 ± 3.58	63.65 ± 6.41	59.40 ± 9.47	41.9 ± 24.7	41.6 ± 24.7
	LH (UI/mL)	25.14 ± 3.10	19.31 ± 1.93	24.33 ± 3.04	23.50 ± 4.37	NA	NA
	E2 (pg/mL)	17.13 ± 2.22	48.08 ± 6.28	17.38 ± 2.61	20.86 ± 7.11	NA	NA
Ovarian reserve parameters (means)	AMH (ng/mL)	0.18 ± 0.04	0.75 ± 0.06	0.15 ± 0.04	0.30 ± 0.05	0.13 ± 0.16	0.18 ± 0.18
	AFC	0	2.33 ± 0.49	0	0	0.5 ± 0.5	1.7 ± 1.4
Pregnancy rate	Spontaneous	16.7% (3/18)		0% (0/12)		7.4% (23/311)	
	After IVF	Not part of the study protocol				4.5% (13/288)	
	Total for the study	10% (3/18)				11.6% (36/311)	

* In this study, the participants who had menstruation restored and decreased FSH were included in the “success group”, while the others were included in the “failure group”. The results of the study were displayed according to this distinction. NA, not available; AFC, antral follicular count; IVF, in vitro fertilization.

**Table 2 jcm-10-05647-t002:** Clinical studies conducted with stem cells to restore fertility in women suffering from spontaneous premature ovarian insufficiency.

Reference	Sample Size	Study Design	Stem Cells Source	Method	Main Outcomes
Edessy M. et al., 2016	*n* = 10	Single-arm study	Autologous bone marrow-derived mesenchymal stem cells	Laparoscopic injection in both ovaries	2/10 (20%): menses resumption 1/10 (10%): pregnancy 1/10 (10%): live birth
Gabr H. et al., 2016	*n* = 30	Single-arm study	Autologous bone-marrow-derived mesenchymal stem cells	Laparoscopic injection in one ovary and one ovarian artery	18/30 (60%): ovulation 1/30 (3.3%): spontaneous pregnancy 3/30 (10%): IVF cycle (no data given by the authors on IVF cycle outcome)
Ding L. et al., 2018	*n* = 14	Randomized, controlled, 2-arm study	Group 1 (*n* = 6): umbilical cord mesenchymal stem cells Group 2 (*n* = 8): umbilical cord mesenchymal stem cells on a collagen scaffold	Transvaginal injection guided by ultrasonography in one ovary	2/14 (14.3%): pregnancy (one in each group): One induced labor for 21-trisomy (collagen group) One ongoing >20 weeks pregnancy
Yan L. et al., 2020	*n* = 61	Single-arm study	Umbilical cord mesenchymal stem cells	Transvaginal injection guided by ultrasonography in both ovaries	4/60 (6.7%): pregnancy (3 with IVF and 1 spontaneous) 4/60 (6.7%): live birth
Mashayekhi M. et al., 2021	*n* = 9	Nonrandomized, controlled, open-label, 3-arm study	Autologous adipose-derived stromal cells; the subjects were divided in 3 groups (*n* = 3 per group) receiving 5, 10 or 15 × 10^6^ cells	Transvaginal injection guided by ultrasonography in one ovary	4/9 (44.4%) had menses restoration: High-dose group: 2/3 Mild-dose group: 1/3 Low-dose group: 1/3 0/9 pregnancy

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
