# Peer review of "Restoration of Fertility in Patients with Spontaneous Premature Ovarian Insufficiency: New Techniques under the Microscope"

_jcm, 2021, doi:10.3390/jcm10235647_

Round 1
Reviewer 1 Report
Review:
Dear Authors,
Thank you for the submission of your article “Restoration of fertility in patients with spontaneous primary ovarian insufficiency: new techniques under the scope”. This article explored possible methods to restore fertility in patients with POI.
Postponing family planning towards and advanced age is a clear trend in our western society. Unfortunately ca 1% of the women by the age of 40 suffer from POI, which makes family planning at a later age difficult or impossible. Up till now no treatment of the fertility aspect of POI was available. With the development of new techniques in IVF and stem cell therapy, we might find new ways to treat these patients. Clinicians need to be aware of the advances regarding these experimental methods
The publication is well organised and interpreted and – in my opinion- worth to be published.
Please allow some remarks/questions, recommendations:
Q1 Abstract: Include a line about unsolved and possible safety issues throuout these procedures
Q2 Page 2 line 45:
“Approaches to preserve fertility in case of iatrogenic POI have been developped and 45 have already shown efficacy”- the word developed is misspelled
Q3 Page 6 Section 3.5. Immuno-modulation in Auto-immune Forms
Q4 Page 6 line 257 “instauration of this immunosuppressor” spelling
Q5 Generally critical approach to the different methods are lacking throughout this manuscript. Especially safety issues should be incorporated at each sections and open issues that remain to be solved before a technique can be considered as a therapy option. Potential adverse outcome like cell proliferation events that occur following stem cell transplantation and may induce malignant formation should be mentioned.
Q6 Please double check the format of the references.
Author Response
Dear Reviewer,
First of all, we take the opportunity of this message to thank you for the time you have spent on reviewing our manuscript and for your valuable comments.
Here below, you may find our answers to each of them.
Q1 Abstract: Include a line about unsolved and possible safety issues throuout these procedures
Authors: In agreement with your comment, we have added the following sentence in the Abstract section: No safety concerns were raised in these studies.
Q2 Page 2 line 45:
“Approaches to preserve fertility in case of iatrogenic POI have been developped and 45 have already shown efficacy”- the word developed is misspelled
Authors: thanks for having found this typo which has been corrected in the manuscript.
Q3 Page 6 Section 3.5. Immuno-modulation in Auto-immune Forms
Authors: thanks for having found this typo which has been corrected in the manuscript.
Q4 Page 6 line 257 “instauration of this immunosuppressor” spelling
Authors: thanks for having found this typo which has been corrected in the manuscript.
Q5 Generally critical approach to the different methods are lacking throughout this manuscript. Especially safety issues should be incorporated at each sections and open issues that remain to be solved before a technique can be considered as a therapy option. Potential adverse outcome like cell proliferation events that occur following stem cell transplantation and may induce malignant formation should be mentioned.
Authors :
We acknowledge your comment and, accordingly, the following text was added in the appropriate sections:
- In section 3.1 “Autologous Platelet-rich Plasma Intraovarian Injection”: No safety concerns were reported in these studies by the authors.
- In section 3.2 “Human Mesenchymal Stem Cells”: No adverse reaction was reported in these four studies.
- In addition, specifically regarding mesenchymal stem cells, the following text has been added in the Discussion section:
No adverse events were reported in any of the studies reviewed in this article. It should however be noted that these techniques use transvaginal or laparoscopic ovarian manipulations which are not devoid of undesirable side effects. In addition, some specific concerns linked to the clinical use of stem cells must be considered. Among them, the possible tumorigenic effect of MSCs is probably the most significant one. Indeed, with their ability to proliferate for a long period of time, their high viability, and their resistance to apoptosis, MSCs could be suspected to behave as tumor cells. In addition, some animal studies also suggest that MSCs could potentiate growth of pre-existing tumor in the host. Beside this oncological concern, other safety issues are suspected: injection of MSCs appears to have pro-thrombotic effects and could also be a vector for prion or viral transmission. The current studies conducted in humans with MSCs are too limited in terms of sample size and duration to properly evaluate the significance of these safety concerns but there is no doubt that the increasing interest around MSC-based therapy will help delineating their exact safety profile.
- In section 3.3 “In-Vitro Activation of Dormant follicles”: No adverse events were reported in these articles.
- In section 3.4 “Mechanical Activation of Dormant Follicles”: No safety issue was reported by the authors
Q6 Please double check the format of the references.
Authors: We thank you for this comment. Accordingly, the references have been corrected to align with the authors recommendations.
We hope that these answers meet your expectations. Thanks again for your review,
Kind regards,
Marie Mawet, Sophie Perrier d’Hauterive, Laurie Henry, Iulia Potorac, Frédéric Kridelka, Michelle Nisolle, Axelle Pintiaux
Reviewer 2 Report
The author made a summary on the current related research about the treatment of POI, which has certain significance for the study in this field. But, many concerns need to be demonstrated.
- In 2016, the European Society for human reproduction and Embryology (ESHRE) published the guideline for POI treatment, changing the full name of POI to "premature ovarian insufficiency (POI). The author should pay attention to the accuracy of the name of the disease.
- The descriptions for the studies are too detailed, not being summarized properly. Under these circumstances, the review is not comprehensive enough and many important studies are missing for a qualified review.
- Nowadays plenty of the treatment options and experimental methods for POI have been studied. For example, in vitro maturation (IVM) method of fertility preservation, in vitro activation of ovarian cortex and autologous transplantation, ovarian tissue cryopreservation, fertoprotective agents, studies about oogonial stem cells (OSC) or artificial gametogenesis etc.. However, this review only provides five aspects of the methods.
- Please provide the schematic of study selection using flow chart in material and method section.
- The manuscript only provides the conventional IVA method, however, a modification technique called drug free IVA have also been reported for POI or POR patients. Please provide more detailed description in results 3.3.
- The references need to be revised or supplemented.
- The style of the reference should be consistent throughout the manuscript.
- Few references were cited in the manuscript but lots of new studies are presented nowadays. Some part of the manuscript only has one reference. Few studies at present or not carefully summarized?
- Some references are out-of-date, such as ref. 1, 27, 28, 29. Please update the references.
- Some references are not appropriate, such as ref. 2, 4, 6, 15, 18.
- The English of your manuscript must be improved before resubmission.
Author Response
Dear Reviewer,
First of all, we would like to thank you for the time you have spent on reviewing and commenting our manuscript. Your comments were very interesting. As you may see below, they were taken into account in our revised version.
1. In 2016, the European Society for human reproduction and Embryology (ESHRE) published the guideline for POI treatment, changing the full name of POI to "premature ovarian insufficiency (POI). The author should pay attention to the accuracy of the name of the disease.
Authors: We thank the reviewer for this comment with which we totally agree. Consequently, the first sentence of the manuscript has been replaced by the following definition:
Premature ovarian insufficiency (POI) has been defined in 2016 by the European Society of Human Reproduction and Embryology (ESHRE) as the onset of menstrual disturbance (oligomenorrhea or amenorrhea) for at least 4 months associated with serum follicle stimulating hormone (FSH) levels ≥ 25 IU/ml measured at least twice with a 4-week interval, before the age of 40 years (reference: Webber L, Davies M, Anderson R, Bartlett J, Braat D, Cartwright B, et al. ESHRE Guideline: management of women with premature ovarian insufficiency. Hum Reprod. 2016;31(5):926-37).
In addition, the term “primary ovarian insufficiency” was replaced everywhere in the manuscript by the term “premature ovarian insufficiency”.
2. The descriptions for the studies are too detailed, not being summarized properly. Under these circumstances, the review is not comprehensive enough and many important studies are missing for a qualified review.
Authors : We agree with the comment of the reviewer. For the sake of clarity, we have therefore summarized the description of all the studies presented in our work. For example, in section 3.1 concerning the intraovarin injection of PRP, we have placed a table which displays the results of the studies. The text was consequently deeply shortened and appears now this way:
Sfakianoudis et al. were pioneers in studying the effects of PRP in women with POI. They initially reported in two articles the cases of four women presenting POI who had rejected the option of oocyte donation. The subjects, aged between 27 and 46 years, were amenorrheic for more than one year (range: 12 to 26 months). They accepted to undergo a transvaginal ultrasound-guided intraovarian injection of autologous PRP. Within 2 months after this procedure, all four women resumed menses and became pregnant: conception was achieved naturally in three of them and using a natural in vitro fertilization (IVF) cycle in one of them. This last woman reported a spontaneous abortion at week 5 of gestation while the three other women completed a healthy complication-free pregnancy.
Following these encouraging results, the same research group conducted a pilot uncontrolled study in which 30 women diagnosed with POI underwent transvaginal intraovarian autologous PRP injection. Recently, Cakiroglu et al. also reported the results of intraovarian injection of autologous PRP in 311 women diagnosed with POI, which currently constitutes the largest study on the topic. The results of these two studies are displayed in Table 1. In both studies, parameters reflecting the activity of the hypothalamic-pituitary-ovarian axis and of the ovarian reserve generally improved, although the results seemed very variable among the participants. Total pregnancy rate in each study was 10.0% and 11.6%, respectively. However, in the second study, a bit more than 30% of the pregnancies were achieved using ovarian stimulation.
Interestingly, in both studies, lower FSH level and higher AMH level prior to treatment were positively correlated with a better outcome of the technique. Therefore, the authors suggested that intraovarian PRP injection helps stimulating existing pre-antral/early antral follicles. The presence of these follicles could thus be a prerequisite that condition the extent of the response.
No safety concerns were reported in these studies by the authors.
Table 1. Clinical studies conducted with intraovarian injection of autologous platelet-rich plasma to restore fertility in women suffering from spontaneous premature ovarian insufficiency.
|
Reference |
Sfakianoudis K. et al., 2020 |
Cakiroglu Y. et al., 2020 |
|||||
|
Number of patients Mean age (years) |
n = 30* |
n = 311 34.8 ±4.3 |
|||||
|
Success group (n=18, 60.0%) 35.11 ±1.57 |
Failure group (n=12, 40.0%) 35.92 ±1.93 |
||||||
|
|
Baseline |
3 months after intraovarian injection of PRP |
Baseline |
3 months after intraovarian injection of PRP |
Baseline |
After intraovarian injection of PRP |
|
|
Mean duration of amenorrhea (months) |
10.06 ±2.62 |
Return of menstruation |
10.17 ±4.76 |
No return of menstruation |
> 4 |
NA |
|
|
Hypothalamic- pituitary-ovarian axis parameters (means) |
FSH (UI/ml)
|
40.61 ±6.05 |
20.67 ±3.58 |
63.65 ±6.41 |
59.40 ±9.47 |
41.9 ±24.7 |
41.6 ±24.7 |
|
LH (UI/ml) |
25.14 ±3.10 |
19.31 ±1.93 |
24.33 ±3.04 |
23.50 ±4.37 |
NA |
NA |
|
|
E2 (pg/ml) |
17.13 ±2.22 |
48.08 ±6.28 |
17.38 ±2.61 |
20.86 ±7.11 |
NA |
NA |
|
|
Ovarian reserve parameters (means) |
AMH (ng/ml) |
0.18 ±0.04 |
0.75 ±0.06 |
0.15 ±0.04 |
0.30 ±0.05 |
0.13 ±0.16 |
0.18 ±0.18 |
|
AFC |
0 |
2.33 ±0.49 |
0 |
0 |
0.5 ±0.5 |
1.7 ±1.4 |
|
|
Pregnancy rate |
Spontaneous |
16.7% (3/18) |
0% (0/12) |
7.4% (23/311) |
|||
|
After IVF |
Not part of the study protocol |
4.5% (13/288) |
|||||
|
Total for the study |
10% (3/18) |
11.6% (36/311) |
|||||
|
* In this study, the participants who had menstruation restoration and decreased FSH were included into the “success group” while the others were included into the “failure group”. The results of the study were displayed according to this distinction. NA, not available; AFC, antral follicular count; IVF, in vitro fertilization |
|||||||
3. Nowadays plenty of the treatment options and experimental methods for POI have been studied. For example, in vitro maturation (IVM) method of fertility preservation, in vitro activation of ovarian cortex and autologous transplantation, ovarian tissue cryopreservation, fertoprotective agents, studies about oogonial stem cells (OSC) or artificial gametogenesis etc.. However, this review only provides five aspects of the methods.
Authors : We recognize the great interest of all the techniques described by the reviewer. These techniques were however essentially developed for fertility preservation in women who are undergoing ovarian damage (mainly for iatrogenic causes such as ovariectomy, chemotherapy and radiations). In this specific context, the main goal is to preserve the existing fertility. Our review focusses on techniques tested to restore fertility in women who suffer from spontaneous premature ovarian insufficiency, i.e. for whom the fertility loss has already been constituted at the time of the diagnosis.
It is true that, with the constant improvement in oncological treatments, incidence of iatrogenic POI has increased these two last decades. Research has persevered in developing strategies to preserve fertility in these women: depending on their age at cancer diagnosis, their genetic profile, the type of oncological treatment, and the time allowed before starting this treatment, the women may be offered different options such as ovarian suppression, ovarian transposition, oocyte and embryo cryopreservation, and ovarian tissue cryopreservation. These approaches have already shown efficacy and there is no doubt that their success will be growing over the years.
In the opposite, restoring fertility in women suffering from spontaneous POI is not achievable and oocytes donation or adoption are generally the only proposals that can be offered to them. As far as fertility is concerned, the big difference between iatrogenic POI and spontaneous POI is that women at risk of developing post-treatment POI may be selected for fertility conservation before the occurrence of POI which is generally not possible for spontaneous POI. Our review attempts to make an overview of different techniques that are tested in these women to restore their ovarian activity.
4. Please provide the schematic of study selection using flow chart in material and method section.
Authors: We thank you for this comment. Accordingly, a flow chart displaying the study selection has been added to the Material and Methods section (Figure 1).
5. The manuscript only provides the conventional IVA method, however, a modification technique called drug free IVA have also been reported for POI or POR patients. Please provide more detailed description in results 3.3.
Authors: The drug-free IVA technique is described in section 3.4 called “Mechanical Activation of Dormant Follicles”, essentially studied by the Zhang and co-workers.
6. The references need to be revised or supplemented.
- The style of the reference should be consistent throughout the manuscript.
Authors: We thank you for this comment. Accordingly, the references have been corrected to align with the authors recommendations.
- Few references were cited in the manuscript but lots of new studies are presented nowadays. Some part of the manuscript only has one reference. Few studies at present or not carefully summarized?
Authors: As mentioned above, our review focusses on the techniques developed during this last decade to restore fertility in women suffering from spontaneous premature ovarian insufficiency. We wanted to limit our research on this specific population, probably currently under-represented in contrast to the numerous studies done in women with iatrogenic POI. As the reviewer has indeed noted, this research represents currently only a few studies (13 articles were retrieved in our current work).
- Some references are out-of-date, such as ref. 1, 27, 28, 29. Please update the references.
Authors: We acknowledge your remark and consequently,
- Reference 1 (Nelson, L.M., Clinical practice. Primary ovarian insufficiency. N Engl J Med, 2009. 360(6): p. 606-14) has been replaced by: Webber L, Davies M, Anderson R, Bartlett J, Braat D, Cartwright B, et al. ESHRE Guideline: management of women with premature ovarian insufficiency. Hum Reprod. 2016;31(5):926-37
- Reference 27 (Kalantaridou, S.N., et al., Treatment of autoimmune premature ovarian failure. Hum Reprod, 1999. 14(7): p. 1777-82) was replaced by: Kirshenbaum, M. and R. Orvieto, Premature ovarian insufficiency (POI) and autoimmunity-an update appraisal. J Assist Reprod Genet, 2019. 36(11): p. 2207-2215.
- Reference 28 (Ferraù, F., et al., Pregnancy after azathioprine therapy for ulcerative colitis in a woman with autoimmune premature ovarian failure and Addison's disease: HLA haplotype characterization. Fertil Steril, 2011. 95(7): p. 2430): This reference was not replaced since it illustrates the rare event where use of an immunosuppressant (other than high doses of corticosteroids) was suspected to have permitted a pregnancy in a woman suffering of auto-immune POI.
- Reference 29 (Nelson, L.M., et al., Development of luteinized graafian follicles in patients with karyotypically normal spontaneous premature ovarian failure. J Clin Endocrinol Metab, 1994. 79(5): p. 1470-5): This reference has also been kept into the manuscript since, to our knowledge, this is the only study which has evaluated the actual monthly ovulation rate in a group of women with spontaneous POI.
- Some references are not appropriate, such as ref. 2, 4, 6, 15, 18.
Authors: In line with your comment,
- References 2, 4 and 6 have been removed from the manuscript.
- Reference 15 (Beer, L., M. Mildner, and H.J. Ankersmit, Cell secretome based drug substances in regenerative medicine: when regulatory affairs meet basic science. Ann Transl Med, 2017. 5(7): p. 170) has been replaced by: Sagaradze, G., et al., Conditioned Medium from Human Mesenchymal Stromal Cells: Towards the Clinical Translation. Int J Mol Sci, 2019. 20(7).
- Reference 18 (Grady, S.T., et al., Effect of intra-ovarian injection of mesenchymal stem cells in aged mares. J Assist Reprod Genet, 2019. 36(3): p. 543-556) was kept since this original work has specifically analysed if the DNA of heterologous stem cells injected into the ovarian tissue is found back in the host cells. Of course, the consequences of such integration, particularly in the host’s oocytes, would have dramatical ethical consequences. Although this single study conducted in mares is not sufficient to totally overcome this potential danger, this is probably one of the best currently available reports on this highly debated topic.
7. The English of your manuscript must be improved before resubmission.
Authors: In line with your comment, our manuscript was entirely reviewed by Dr Adrian Daly who is Native English Speaker.
We sincerely hope that this revised version of the manuscript will meet your expactations.
Thanks again for your valuable review,
Sincerely,
Marie Mawet, Sophie Perrier d’Hauterive, Laurie Henry, Iulia Potorac, Frédéric Kridelka, Michelle Nisolle and Axelle Pintiaux
Round 2
Reviewer 2 Report
I am satisfied with the revision.